# Effects of Sulfur Levels in Fermented Total Mixed Ration Containing Fresh Cassava Root on Feed Utilization, Rumen Characteristics, Microbial Protein Synthesis, and Blood Metabolites in Thai Native Beef Cattle

**DOI:** 10.3390/ani9050261

**Published:** 2019-05-21

**Authors:** Chanadol Supapong, Anusorn Cherdthong, Metha Wanapat, Pin Chanjula, Sutipong Uriyapongson

**Affiliations:** 1Tropical Feed Resources Research and Development Center (TROFREC), Department of Animal Science, Faculty of Agriculture, Khon Kaen University, Khon Kaen 40002, Thailand; chanadol@kkumail.com (C.S.); metha@kku.ac.th (M.W.); suthipng@kku.ac.th (S.U.); 2Department of Animal Science, Faculty of Natural Resources, Prince of Songkla University, Songkhla 90112, Thailand; pin.c@psu.ac.th

**Keywords:** energy source, mineral, rumen fermentation, ruminant, tropical feed

## Abstract

**Simple Summary:**

Feeding of fresh cassava root to animals is restricted because it contains hydrocyanic acid at a high level, which is the origin for poisoning. High levels of hydrocyanic acid from fresh cassava root could be detoxified by sulfur addition to become nontoxic to cattle. The addition of 2% sulfur in a fermented total mixed ration containing fresh cassava root and ensiling for 7 days could improve dry matter digestibility, efficiency of microbial protein synthesis, and concentrations of total volatile fatty acid, propionic acid, and blood thiocyanate.

**Abstract:**

The influence of sulfur included in fermented total mixed ration (FTMR) containing fresh cassava root on rumen characteristics, microbial protein synthesis, and blood metabolites in cattle was evaluated. Four Thai native beef cattle were randomly assigned according to a 2 × 2 factorial in a 4 × 4 Latin square design, and dietary treatments were as follows: factor A included a level of sulfur at 1% and 2% in total mixed ration, and factor B featured ensiling times at zero and 7 days. Digestibility of dry matter was increased when FTMR was supplemented with 2% sulfur. Blood thiocyanate increased by 69.5% when ensiling time was 7 days compared to no ensiling (*p* < 0.01). Bacterial populations were significantly different in the FTMR containing sulfur at 2% and 7 days of ensiling. Furthermore, microbial crude protein and efficiency of microbial protein synthesis were higher in the FTMR containing 2% sulfur and 7 days of ensiling (*p* < 0.01). Thus, high levels of hydrocyanic acid from fresh cassava root could be detoxified by a sulfur addition with an ensiling process to become nontoxic to cattle.

## 1. Introduction 

Cassava root is the main carbohydrate source produced for ruminant consumption in the tropical zone, particularly Thailand. The starchy root is produced generally for human consumption or as a carbohydrate source for animal livestock feed [1]. Fresh cassava root as an energy supplement is beneficial in ruminant diets because of its low price and reduced processing, as compared to cassava chips, as well as its convenience for the farmer [2]. Moreover, the cassava chips product has problems in the rainy season for its sun-drying process. However, feeding of fresh cassava root to animals is restricted because it contains hydrocyanic acid (HCN) at a high level, which is the origin for poisoning. Fresh cassava root contains around 90–114 mg/kg of HCN [3,4].

Fermented total mixed ration (FTMR) is a basic process to potentially maintain nutrient utilization, extend preservation, and reduce antinutritive substances in feeds [5]. FTMR is roughage of a concentrated mixture that is fermented under anaerobic conditions (i.e., ensiling), and this process may affect the reduction of antinutritive substances such as mimosin or HCN. Nkosi and Meeske [6] studied FTMR, adding *Lactobacillus buchneri* and fermenting for 90 day, and revealed an increased feed intake and growth rate in sheep when compared to those fed the nonfermented feed. In addition, Vasupen et al. [7] indicated that FTMR can improve the digestibility of nutrients and milk production in lactating dairy cows. Thus, feeding FTMR not only reduced HCN in fresh cassava root but also may have the potential to improve feed utilization and animal production.

Ruminants receiving HCN from crops could eliminate it by enzymes such as rhodanese and β mercapto pyruvate sulfur transferase, which are excreted from rumen microbes [8]. Rhodanese is a sulfur transferase that catalyzes the deformation of HCN and thiosulfate (or other suitable sulfur donors) to the less toxic thiocyanate, which is removed in the urine. The main sources of sulfur for HCN elimination are sulfur amino acids, cysteine and methionine, or elemental sulfur. Cherdthong et al. [4] reported that supplementing fresh cassava root at 1.5% body weight (BW) with a feed block containing 4% sulfur did not adversely affect roughage intake, rumen fermentation, nor blood urea-nitrogen (BUN), whereas HCN was reduced. Furthermore, an in vitro study demonstrated that the inclusion of 2% sulfur in TMR containing fresh cassava root fermented for 21 day could improve kinetics of gas and nutrient digestibility while maintaining ruminal fermentation parameters and 54% rate of HCN disappearance [9].

It was hypothesized that sulfur addition and the ensiling method could improve feed utilization and rumen fermentation efficiency. Thus, the goal of this work was to study the influence of sulfur doses included in FTMR containing fresh cassava root on nutrient intake, rumen characteristics, microbial protein synthesis, and blood metabolites in Thai native breeds of cattle.

## 2. Materials and Methods

All cattle were handled per the guidelines of the Committee of Animal Experimentation on Animal Welfare for Experimental Animals of Khon Kaen University (record no. ACUC-KKU 32/61), and all experiments were performed under the Ethics of Animal Experimentation of the National Research Council of Thailand.

### 2.1. Feeding of Animals and Experimental Design

Four Thai native beef cattle with an initial BW of 90 ± 5.0 kg were randomly assigned according to a 2 × 2 factorial in a 4 × 4 Latin square design, and dietary treatments were as follows: factor A included a level of sulfur supplementation at 1% and 2% in total mixed ration (TMR), and factor B featured ensiling times at 0 and 7 day. The optimum sulfur level and ensiling time were selected from our previous study [9], which indicated that it could improve the kinetics of gas and nutrient digestibility while maintaining ruminal fermentation parameters and the rate of HCN disappearance.

Fresh cassava root was collected from the Khon Kaen province areas of Thailand and then was chopped into chip form to go through a sieve (1 cm) by a machine (Yasothon Thailand). The FTMR was a rice straw chop with the concentrate mixed by machine and then adjusted with water to a moisture content of 55%, and lactic acid bacteria were not inoculated to the mixture before ensiling. FTMR then went through ensiling under anaerobic conditions in a plastic, sealed container to analyze ingredient and chemical composition (Table 1). Samples were stored outdoors (25–32 °C) for 7 day. All cattle were kept in individual pens, were fed FTMR on an ad libitum basis, were offered it two times per day at 07:00 and 16:00, and had water accessible at all times. Research proceeded for four, 21 day periods, beginning with a 14 day preliminary adjustment period. Each period ended with 7 day in which the animals were transferred to metabolism crates and fed FTMR at 90% of the previous feed intake, during which all feces and urine were sampled. The feed intake was previously measured. Feces and urine samples were used for nutrient digestibility and microbial protein synthesis.

### 2.2. Sample Collection and Chemical Analysis

During the last 7 day of each period, samples of FTMR, feces, and urine were taken by using a total collection method, as animals were on metabolism crates to study nutrient digestibility and nitrogen balance. FTMR and feces were dried at 60 °C for 48 h then ground to pass through a 1 mm screen (Cyclotech Mill, Tecator, Hoganas, Sweden) and used for chemical analysis. The diets and feces were chemically analyzed for dry matter (DM), ash and crude protein (CP), and acid detergent fiber (ADF) [10]. Neutral detergent fiber (NDF) was measured by the procedure of Van Soest et al. [11]. HCN concentrations in the FTMR and fresh cassava root were estimated according to Bradbury et al. [12]. At a 510 nm wavelength, an absorbent picrate solution was recorded by spectrophotometry and estimated total cyanide concentrations as (mg/kg)  =  396  ×  absorbance reading. Prechlorination eliminated all cyanide amenable to chlorination, as performed by Franson [13]. Digestible organic matter fermented in the rumen (DOMR) and digestible organic matter intake (DOMI) were calculated following the method explained by Agricultural Research Council (ARC) [14] and Kearl [15].

Nitrogen (N) was analyzed from the urine samples, and N utilization was calculated according to the equation described by the AOAC [10]. Allantoin contained in the urine was evaluated by the procedure of Chen et al. [16] using high-performance liquid chromatography (HPLC). The concentration of microbial purine absorbed (*X* mmol/day) corresponding to the purine derivatives excreted (*Y* mmol/day) was estimated based on the relationship derived by Chen and Gomes [17]: *Y* = 0.85*X* + (0.385*W*^0.75^), where *Y* is the excretion of purine derivatives (mmol/day), *X* the microbial purines absorbed (mmol/day), and *W* is body weight of the animal (g/kg BW^0.75^).

The supplies of microbial crude protein (MCP) in grams per day were calculated as follows:$$\mathrm{MCP}\;(g/\mathrm{day})\;=\;\frac{Xx\;70}{0.116\;x\;0.83\;x\;1000}\;=\;0.727\;x\;X,$$
where *X* is the absorption of purine derivatives in mmol/day, which followed the assumptions calculated by Chen and Gomes [17]. The digestibility of microbial purine was 0.83, and the N concentration of purine was 70 mg N/mmol. The ratio of purine-N to total N in mixed rumen microbes was 11.6:100.

Ten milliliters of blood was sampled from the animals’ jugular vein at 0 and 4 h after feeding on the 21st day of each period to analyze BUN and blood thiocyanate, as described by Lambert et al. [18]. Approximately 45 mL of ruminal fluid samples were collected (also at 0 and 4 h after feeding) on the 21st day of each period through a stomach tube connected to a vacuum pump. Ruminal fluid pH and temperature were immediately determined using a portable pH and temperature meter (HANNA Instruments HI 8424 microcomputer, Singapore). Ruminal fluid was mixed with sulfuric acid (H_2_SO_4_) at a 1:9 ratio, centrifuged at 16,000× *g* for 15 min, and then analyzed for ammonia nitrogen (NH_3_-N) (Kjeltech Auto 1030 analyzer, Tecator, Hoganiis, Sweden), while total volatile fatty acid (VFA) and VFA profiles were performed using HPLC (instruments by controller water model 600E, Water model 484 UV detector, column Novapak C18, column size 4 × 150 mm^2^, mobile phase 10 mM H_2_PO_4_ (pH 2.5); ETL Testing Laboratory, Inc., Cortland, NY, USA) according to Samuel et al. [19]. 

One milliliter of ruminal fluid was mixed with 9 mL of formalin solution, and then its population of bacteria and protozoa was counted using a microscope with a hemocytometer (Boeco, Hamburg, Germany). Blood metabolites and rumen characteristics were expressed as mean values.

### 2.3. Calculations and Statistical Analysis

Data appropriate for the 2 × 2 factorial in a 4 × 4 Latin square design were analyzed using a Generalized Linear Model (GLM) procedure (SAS, 1989). The model is as follows:*Y_ijk_ = μ + M_i_ + E_j_ + A_k_ + P_l_ + ε_ijk_*
where *Y_ijk_* is the observation for cattle *j* receiving diet *i* in period *k*; *μ* is the overall mean; *M_i_* is the effect of the various doses sulfur (*i* = 1, 2); *E_j_* is the effect of the different ensiling times (*j* = 0,7); *A_k_* is the effect of the animal (*k* = 1, 2, 3, 4); Pl is the effect of the period (*l* =1, 2, 3, 4); and *ε_ijk_* is the residual effect. Results were presented as mean values with the standard error of the means. Differences between treatment means were tested by Duncan’s new multiple range test, and differences amongst means were considered statistically significant at *p* < 0.05.

## 3. Results and Discussion

### 3.1. Nutritional Contents in the Diets

The chemical compositions of the treatments are shown in Table 1. FTMR consisted of CP, NDF, and ADF at 11.6% to 12.8% DM, 56.5% to 59.0% DM, and 22.7% to 23.4% DM, respectively. The composition of the fresh cassava root consisted of CP, NDF, and ADF at 2.08% DM, 53.13% DM, and 31.19% DM, respectively. Fresh cassava root contained 110 ppm of HCN; however, the chemical composition of the fresh cassava root might vary depending on factors such as variety, soil fertility, state of growth, etc. The pH was lowest when supplemented with 2% sulfur levels. In addition, the ensiling time of 7 day also reduced the pH. This could be due to an increasing number of lactic acid bacteria during the ensiling process, which results in a dominant population of epiphytic micro-organisms in fermented feedstuff and promotes lactic acid fermentation to restrain a generation of undesired microbes. Lactic acid bacteria produced sufficient lactic acid and, thus, decreased pH when the ensiling time was extended for 7 day. The pH was one major part that affected the extent of fermentation and quality of silage, as a high lactic acid amount was essential for storage [20]. Even if the pH values of all FTMR with or without sulfur levels were not below 4.2, they were well-preserved in the anaerobic condition [21]. As previously reported, an in vitro study noted that pH in the FTMR ranged from 6.00 to 6.70 when ensiling for 7 day [9].

The HCN concentrations of FTMR supplementations at 1% and 2% sulfur were reduced 99.3% to 99.5% when compared to the fresh cassava root. Furthermore, reduction of the HCN concentration, even without the ensiling process, could be due to the preparation of the TMR process by machine, which may generate high temperatures, thereby resulting in lower HCN. Furthermore, FTMR supplementation at 2% sulfur and 7 d ensiling time was lowest when compared with other groups. Sulfur is an essential mineral for cell synthesis and maintenance of the cellular metabolisms of microbes in the rumen [22]. In addition, sulfur can detoxify HCN in feedstuff to another substance that is not toxic to the animals [8]; this mode of action will be provided later on. Furthermore, it could be due to the fact that a long ensiling time can provide high temperature conditions and acid from microbial activities, thus detoxifying the HCN in FTMR. Microorganisms can break down toxins into organic acids and inactive laminarase enzymes. As Kimaryo and Massawe [23] reported, during the fermentation of fresh cassava root, the HCN was reduced from 176.3 to 8.2 ppm after 5 day of ensiling. Boonnop et al. [24] studied fermented fresh cassava products and showed that the products had a low HCN content of 0.5 ppm after 6 day of fermentation. In current ensiling, microorganisms convert glucose to organic acids that cause the pH to decrease. The lower pH can reduce laminarase enzyme activity to degrade linamarine to HCN; therefore, supplementation of sulfur and the ensiling process could contribute to decreasing HCN content [25].

### 3.2. Feed Utilization

The DM intake and nutrient digestibility of FTMR in animals are shown in Table 2. There was no significance (*p* > 0.05) between sulfur levels and ensiling times on all parameters. Total FTMR intakes were similar among feed groups and ranged from 100.8 to 109.1 g/kg BW^0.75^. The levels of sulfur supplementation did not differ in the intake of nutrients and energy (*p* > 0.05). Differences in ensiling times did not affect nutrient digestibility. Nevertheless, the digestibility of DM was increased when supplemented at 2% sulfur (*p* < 0.05) in the FTMR, which was similar to the report by Supapong and Cherdthong [9], who indicated that a 2% sulfur level in the FTMR increased DM digestibility. This might be due to increasing numbers of rumen bacteria (Table 3) in cattle fed with FTMR containing a sulfur level at 2% DM, as compared to 1% sulfur diets, leading to increased DM digestibility. In ruminants, sulfur is absolutely needed to improve ruminal microorganism growth and leads to an improved digestibility coefficient [26]. Similar results were reported by Promkot and Wanapat [27], who found that cows fed 0.4% sulfur had an increased fiber digestibility over those fed 0.15% sulfur in their diet.

### 3.3. Ruminal Fermentation, Blood Metabolites, Microorganisms, and Purine Derivatives

The feeding of FTMR containing fresh cassava root did not change ruminal pH and temperature (Table 3; *p* > 0.05), which ranged from 6.9 to 7.1 and 38.7 °C to 39.1 °C, respectively. Ruminal pH and temperature values for all FTMR were in the suitable range for microbial synthesis and activity to ferment the feed in the rumen [28]. The concentration of ruminal NH_3_-N values showed no interactions between sulfur levels and ensiling times. Ruminal NH_3_-N was significantly lowest at 15.2 mg/dL when supplemented with sulfur at a 2% level (*p* < 0.02). This could be due to ruminal microorganisms using NH_3_-N as material for cell synthesis and carbohydrate as an energy source. Thus, the utilization of NH_3_-N with the provided sulfur for microbial protein synthesis might be expected [22]. 

BUN was not altered with the dietary treatments (*p* > 0.05), and BUN concentration was in the optimal range for balanced protein utilization in the animals. This could be because ruminal NH_3_-N is highly beneficial for microbial protein synthesis, along with carbohydrates from fresh cassava root and sulfur, which thereby resulted in an unaltered BUN level. This result was related to nitrogen utilization and microbial protein synthesis. The concentration of blood thiocyanate were not altered by the levels of sulfur and ensiling times (*p* > 0.05). Blood thiocyanate increased by 69.5% with an ensiling time of 7 day compared to no ensiling (*p* < 0.01), possibly because the ensiling time can provide high-temperature conditions and acid from microbial activities to detoxify the HCN in FTMR. In addition, the ensiling time could change the structure of sulfur to sulfate and sulfide, which combines with thiosulfate to modulate to thiocyanate [29]. NRC [30] reported that the animals that received HCN in diets (such as fresh cassava root or sorghum) may need a high addition of sulfur to stimulate sulfur for elimination into the thiocyanate form. HCN can be eliminated by rhodanese and β-mercaptopyruvate-sulfurtransferase through rumen microbes [8]. Rhodanese is an enzyme that contains sulfur and can change the form of HCN into thiocyanate, which is transferred out in the urine. This result agreed with the research of Onwuka et al. [31], who also revealed that the level of sulfur in the feed correlated with blood thiocyanate when a dried cassava-based diet was fed to sheep. In addition, inclusion of a 1% sulfur level in the diet can also reduce HCN and convert it to thiocyanate. Furthermore, adding fresh cassava foliage with 0.5% sulfur reduced HCN concentration, while the percentage of thiocyanide concentration during the in vitro fermentation increased [32].

The sulfur levels and ensiling time did not affect the protozoal concentration (*p* > 0.05). Bacterial populations were significantly different with an FTMR inclusion of sulfur at 2% and 7 day of ensiling, which increased by 5.7% when compared with the FTMR inclusion of 1% sulfur and no ensiling (*p* < 0.01). This may be because ensiling time provides suitable fermentable energy for ruminal microbial synthesis and can stimulate feed digestion, as found in previous studies [33,34]. In addition, it probably is because the animal relies on rumen microbes to change sulfate to hydrogen sulfite, which is supplied to produce methionine and cysteine for microorganism synthesis [4]. As with sulfur-deficient diets, sulfur supplementation improves performance by enhancing bacterial protein synthesis in the rumen and improving the amino acid balance [35]. 

Table 4 shows the data on N balance and purine derivatives in animals fed FTMR containing fresh cassava root. Total N intake and N excretion were not altered by diets (*p* > 0.05) and ranged from 76.7 to 79.5 g/day and 23.8 to 25.8 g/day, respectively. Total N intakes were similar in all treatments, as all Thai native beef cattle were fed the diet ad libitum. Therefore, all animals consumed the same amount of total dry matter intake (DMI), which showed no difference on total N intake. The results on purine absorption were significantly different among the groups, which showed increase (125.9 to 130.4 mmol/day, *p* < 0.01) in the steers supplemented with FTMR ensiled for 7 day and had no effect on sulfur levels (*p* > 0.05). Furthermore, MCP and the efficiency of microbial protein synthesis (EMPS) were different among treatments, and higher values were found in the groups supplemented with FTMR that contained 2% sulfur and was ensiled for 7 day, while the lowest was found in the non-ensiling FTMR diet (*p* < 0.01). Similarly, Khosravi et al. [36] revealed that an ensiling time could increase the rate of nonstructural carbohydrate degradation, which could supply a substrate for microorganism growth [37]. It can also be noted that the microorganisms in the rumen synthesize sulfur containing amino acids and aid in the production of MCP.

### 3.4. Ruminal Volatile Fatty Acid (VFA) Concentration

The proportions of total VFA, propionic acid, and butyric acid were significantly different among FTMR diets (*p* < 0.05) (Table 5). The amounts of total VFA, acetic acid, propionic acid, and butyric acid ranged from 105.8 to 113.8 mmol/L, 64.6 to 66.6, 21.6 to 27.3, and 8.1 to 11.8 mol/100 mol, respectively. The interaction between sulfur and ensiling time on total VFA and propionic acid was observed (*p* < 0.01). A diet with an exceedingly high level of sulfur (2% of the DM) with an ensiling time of 7 day increased total VFA and propionic acid concentrations. This could be because ensiling time improves the amount of available carbohydrates, which are rapidly fermented into total VFA and propionate in the rumen. In addition, a long ensiling time could provide more carbohydrates and result in high total VFA and propionic acid concentrations in the rumen [38]. Furthermore, when a high amount of sulfur was supplemented, the increase in the concentration of ruminal propionate may also note that the concentration of propionate can be used as a sink for hydrogen sulfide when excess ruminal available sulfur is offered [39]. Similarly, Promkot et al. [32] revealed that an increase in total VFA and propionate were noted with 1% sulfur supplemented in fresh cassava foliage, as sulfur is a precursor to microbial protein synthesis in the rumen. Furthermore, Supapong and Cherdthong [9] found that supplementation of 2% sulfur in TMR containing fresh cassava root increased propionic acid by 10.9% when compared to an absence of sulfur supplementation.

## 4. Conclusions

The addition of 2% sulfur in FTMR containing fresh cassava root and ensiling for 7 days could improve dry matter digestibility, efficiency of microbial protein synthesis, and concentrations of total volatile fatty acid, propionic acid, and blood thiocyanate. In addition, the high levels of HCN from fresh cassava root could be detoxified by sulfur addition with an ensiling process to become nontoxic to cattle. However, these findings should be further investigated regarding milk production to elucidate the effect of FTMR on animal milk production.

## Figures and Tables

**Table 1 animals-09-00261-t001:** Ingredients and chemical composition of fermented total mixed ration (FTMR) used in the experiment.

Item	1% Sulfur		2% Sulfur		Fresh Cassava Root
	0 Day Ensiling	7 Days Ensiling	0 Day Ensiling	7 Days Ensiling	
Ingredients, %DM					
Rice straw	40.0	40.0	40.0	40.0	
Fresh cassava root	40.0	40.0	40.0	40.0	
Soybean meal	5.0	5.0	5.0	5.0	
Palm kernel meal	4.0	4.0	3.0	3.0	
Rice bran	3.0	3.0	3.0	3.0	
Urea	2.0	2.0	2.0	2.0	
Pure sulfur	1.0	1.0	2.0	2.0	
Mineral premix	1.0	1.0	1.0	1.0	
Molasses, liquid	3.0	3.0	3.0	3.0	
Salt	1.0	1.0	1.0	1.0	
Chemical composition					
Dry matter, %	55.1	55.8	55.0	55.3	32.0
Organic matter, %DM	95.4	95.5	95.7	96.0	92.6
Ash, %DM	4.6	4.5	4.3	4.0	7.4
Crude protein, %DM	12.6	12.8	11.6	11.7	2.1
Neutral detergent fiber, %DM	56.5	58.1	58.3	59.0	53.1
Acid detergent fiber, %DM	22.7	23.4	22.9	22.7	31.2
pH	6.11	6.08	5.11	4.27	
HCN, ppm	0.76	0.70	0.76	0.44	110.00

DM: Dry matter, and HCN: hydrocyanic acid.

**Table 2 animals-09-00261-t002:** Influence of FTMR on dry matter intake, nutrient intake, and digestibility coefficients in Thai native beef cattle.

Item	1% Sulfur		2% Sulfur		SEM	*p*-Value		
	0 Day Ensiling	7 Days Ensiling	0 Day Ensiling	7 Days Ensiling		S	E	S × E
Dry matter intake								
% BW	3.5	3.6	3.3	3.5	0.38	0.39	0.39	0.82
g/kg BW^0.75^	106.2	100.8	107.3	109.1	2.53	0.47	0.79	0.58
Nutrient intake, kg/d								
Dry matter	3.7	3.8	3.7	3.8	0.61	0.93	0.81	0.94
Organic matter	3.5	3.4	3.5	3.5	0.59	0.82	0.92	0.98
Crude protein	0.5	0.5	0.5	0.5	0.22	0.84	0.88	0.99
Neutral detergent fiber	0.7	0.7	0.7	0.7	0.27	0.79	0.79	0.89
Acid detergent fiber	0.2	0.3	0.3	0.3	0.17	0.42	0.75	0.67
Estimated energy intake								
DOMI ^c^, kg/d	2.3	2.3	2.3	2.3	0.47	0.88	0.96	0.96
DOMR ^d^, kg/d	1.5	1.5	1.5	1.5	0.37	0.93	0.79	0.93
ME, MJ/d	8.8	8.6	8.8	8.7	0.92	0.92	0.87	0.99
ME, MJ/kg DM	2.6	2.5	2.6	2.5	0.45	0.95	0.76	0.95
Digestibility coefficients								
Dry matter, %	0.71 ^a^	0.71 ^a^	0.74 ^b^	0.73 ^b^	0.09	0.03	0.77	0.55
Organic matter, %DM	0.74	0.75	0.75	0.75	0.07	0.78	0.41	0.41
Crude protein, %DM	0.67	0.68	0.68	0.69	0.15	0.70	0.48	0.96
Neutral detergent fiber, %DM	0.65	0.64	0.65	0.65	0.14	0.73	1.00	0.81
Acid detergent fiber, %DM	0.39	0.38	0.40	0.41	0.13	0.29	0.76	0.44

S: *p*-value level of sulfur in the diet. E: *p*-value ensiling times in the diet. SEM: standard error of mean. ^a,b^ The mean was significantly different within rows (*p* < 0.05). BW: Body weight of Thai native beef cattle. ^c^ DOMI: Digestible organic matter intake. ^d^ DOMR: Digestible organic matter fermented in the rumen. ME: Metabolizable energy.

**Table 3 animals-09-00261-t003:** Effects of FTMR on rumen ecology, microorganisms, blood urea-nitrogen, and blood thiocyanate in Thai native beef cattle.

Item	1% Sulfur		2% Sulfur		SEM	*p*-Value		
	0 Day Ensiling	7 Days Ensiling	0 Day Ensiling	7 Days Ensiling		S	E	S × E
Rumen ecology								
Ruminal pH	7.1	6.9	7.1	7.0	0.27	0.88	0.14	0.47
Ruminal temperature, °C	39.1	39.0	39.1	38.7	0.28	0.24	0.21	0.12
NH_3_-N concentration, mg/dL	21.0 ^a^	18.7 ^a^	16.3 ^b^	15.2 ^b^	1.21	0.02	0.26	0.70
Ruminal microbes, cell/mL								
Protozoa, ×10^6^	10.3	10.4	10.9	10.1	0.56	0.63	0.30	0.23
Bacteria, ×10^9^	35.1 ^a^	36.3 ^b^	35.8 ^b^	37.1 ^c^	0.14	0.04	0.01	0.70
Blood parameters								
Blood urea-N concentration, mg/dL	9.9	9.0	8.6	9.0	1.63	0.85	0.96	0.78
Blood thiocyanate concentration, mg/dL	10.8 ^a^	16.6 ^b^	13.3 ^a^	18.1 ^b^	1.36	0.26	0.01	0.78

S: *p*-value level of sulfur in diet. E: *p*-value ensiling times in diet. SEM: standard error of mean. ^a,b,c^ the mean was significantly different within rows (*p* < 0.05).

**Table 4 animals-09-00261-t004:** Effect of FTMR supplementation on nitrogen (N) balance and purine derivative.

Item	1% Sulfur		2% Sulfur		SEM	*p*-Value		
	0 Day Ensiling	7 Days Ensiling	0 Day Ensiling	7 Days Ensiling		S	E	S × E
N balance, g/d								
N intake	77.8	76.7	79.5	78.5	2.79	0.83	0.89	0.10
N excretion	25.8	24.1	25.2	23.8	1.19	0.76	0.29	0.10
Fecal excretion								
Output, kg/d	1.2	1.2	1.1	1.1	0.35	0.63	0.77	0.77
Total N, g/d	5.9	6.8	6.1	6.7	1.02	0.96	0.49	0.93
Total N, %N excretion	23.3	29.7	24.5	28.2	2.25	0.98	0.34	0.79
Urinary excretion								
Output, L/d	1.3	1.6	1.3	1.5	0.41	0.61	0.14	0.61
Total N, g/d	19.9	17.3	19.1	17.1	2.13	0.82	0.30	0.90
Total N, % N excretion	76.7	70.3	75.5	71.9	2.25	0.98	0.34	0.79
N absorption	71.9	69.9	73.5	71.8	2.81	0.83	0.82	0.99
N retention	52.0	52.7	54.3	54.7	2.62	0.76	0.94	0.98
% of N retention to N intake	66.5	68.3	67.5	69.0	1.50	0.70	0.48	0.96
Purine derivative, mmol/day								
Excretion	79.5 ^a^	136.2 ^c^	101.1 ^b^	140.1 ^c^	3.56	0.34	0.01	0.50
Absorption	59.1 ^a^	125.9 ^c^	84.5 ^b^	130.4 ^c^	3.86	0.34	0.01	0.50
MCP, g/d	271.5 ^a^	465.3 ^c^	345.2 ^b^	478.5 ^c^	6.58	0.34	0.01	0.50
MCP (g/ digest OM kg)	119.5 ^a^	212.0 ^c^	156.5 ^b^	211.6 ^c^	5.11	0.50	0.02	0.49
EMPS	18.4 ^a^	32.6 ^c^	24.1 ^b^	32.5 ^c^	2.00	0.50	0.02	0.49

S: *p*-value level of sulfur in diet. E: *p* -value ensiling times in diet. SEM: standard error of mean. ^a,b,c^ The mean was significantly different within rows (*p* < 0.05). MCP: microbial crude protein. EMPS: efficiency of microbial protein synthesis. N: Nitrogen.

**Table 5 animals-09-00261-t005:** Volatile fatty acid (VFA) of rumen fluid of Thai native beef cattle fed FTMR.

Item	1% Sulfur		2% Sulfur		SEM	*p*-Value		
	0 Day Ensiling	7 Days Ensiling	0 Day Ensiling	7 Days Ensiling		S	E	S × E
Total VFA, mmol/L	105.8 ^a^	110.3 ^b^	113.6 ^c^	113.8 ^c^	0.79	0.01	0.01	0.01
VFA profiles, mol/100 mol								
Acetic acid	66.6	66.3	65.0	64.6	0.47	0.29	0.10	0.82
Propionic acid	21.6 ^a^	22.9 ^a^	25.3 ^b^	27.3 ^c^	0.29	0.01	0.01	0.01
Butyric acid	11.8 ^c^	10.9 ^c^	9.7 ^b^	8.1 ^a^	0.43	0.01	0.01	0.08

S: *p*-value level of sulfur in diet. E: *p*-value ensiling times in diet. SEM: standard error of mean. ^a,b,c^ The mean significantly differed within rows (*p* < 0.05). VFA: Volatile fatty acid.

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
