# Peer review of "Effects of Sulfur Levels in Fermented Total Mixed Ration Containing Fresh Cassava Root on Feed Utilization, Rumen Characteristics, Microbial Protein Synthesis, and Blood Metabolites in Thai Native Beef Cattle"

_animals, 2019, doi:10.3390/ani9050261_

Round 1
Reviewer 1 Report
I reviewed the manuscript animals-490518, titled “Effect of sulfur levels in fermented total mixed ration containing fresh cassava root on feed utilization, microbial protein synthesis and blood metabolites in Thai native beef cattle”.
The aim of the work was to study the influence of 2 different doses of sulfur included in FTMR containing cassava root, on nutrient intake, rumen characteristics, microbial protein synthesis and blood metabolites in Thai native beef cattle.
I think that the experimental design and the parameters analyzed were chosen properly to answer the research aim, and some results are interesting. However, considering the literature presented also by the authors, I don’t see a high novelty feature in this research.
The preparation of the paper was not very well done, there are some inconsistencies, missing parts, not always clear result explanation.
Please, following some comments on the different sections, and few detailed comments referring to specific lines.
Title: the title, although long, doesn’t include all the parameters analyzed; I assume that feed intake and digestibility are included in “feed utilization”, however is totally missing the analysis of rumen characteristics. I suggest to modify the title.
Simple summary: There is space to improve it (max 200 words); I think is not very complete and clear, I suggest to improve it. As example, in this sentence “A high levels of HCN from fresh cassava root could be detoxified by a sulfur addition with an ensiling process to become nontoxic to cattle” the concept of detoxification is repeated twice, and it seems that the ensiling process is needed to achieve it (while, I don’t think it’s true).
Abstract: it is too long, the Instructions for the Authors clearly set a limit of 200 words, while the present abstract is of about 400. It needs to be reduced.
Introduction: generally, is complete and give an overall background on the topic. I have a specific remark on the hypothesis: from the introduction is clear that both the sulfur addition and the ensiling process reduce the HCN concentration in cassava root, so I would remove this from the hypothesis, and leave only the effect on feed utilization and rumen fermentation efficiency.
I also suggest to specify in the goal of the work that the breed chosen was a Thai native breed of cattle.
Materials and methods:
Something very important, the Statistical analysis is totally missing, please provide information about this.
L.103: The description of the sampling protocol is very poor, please improve (e.g. were samples collected every day? Once a day? In the tables there is reference at 2 sampling times, but here nothing. Amount of sampling?)
L.85-86: the authors correctly state that the decision of the ensiling time was based on previous research, however they should do it also for the sulfur concentration, on what basis they choose 1 and 2%?
L.89: the moisture content can’t be expressed as % of DM, considering that the DM doesn’t include the water.
L.90: what does it mean that FTMR was ensiled in plastic sealed containers to “ensure ingredient and chemical composition”? Please, make it more clear.
L.95: previously the animals were fed ad libitum, while in this phase at 90% of the previous feed intake; please, mention that the feed intake was previously measured.
L.106: remove the organic matter from the chemical analysis; organic matter is obtained by calculation, not by chemical analysis.
L.108: Please correct “Ata510”
L.107,114: I don’t recommend to start a sentence with an abbreviation or symbol.
L.120: specify what MCP is.
L.134: Please, give more details on the protocols used, and some reference.
Results and discussion:
without having the description of the statistical analysis is not very easy to understand and comment the results. In the tables the authors present the data at 0 and 4h after feeding, and the mean of these two sampling times, however in the materials and methods there is no description of this sampling protocol, and in the description of the results they don’t compare the two sampling times, but only the mean value. Please, justify this decision, and if they are not needed, I suggest to remove the values of the two sampling times from the tables.
Moreover, although for most of the parameters measured there is not effect of the interaction by the two variables (S*E), often the authors compare the values obtained with the 2% sulfur after 7d of ensiling with those obtained with the 1% sulfur at 0d of ensiling process. I believe this is incorrect, and can create confusion in understanding the results. I suggest to modify this approach.
L. 139-140: simplify this first sentence, a suggestion: “The chemical composition of the treatments is shown in Table 1”.
I would separate the pH and the HCN from the chemical composition description.
L. 146: were lactic acid bacteria inoculated to the mixture before ensiling? Specify it in the materials and methods sections. The authors suggest that the lowest pH for the 2% sulfur treatment after 7 days of ensiling is due to the bacteria action during the ensiling process; this might be true, however why it’s not so for the 1% sulfur treatment?
L.152-153: I don’t understand this sentence: “…, and it is attributed that this critical pH value varies with the DM content of the feed”; please clarify.
L.155-156: Also the other treatments reduced the HCN concentration more than 99% even without the ensiling process, why the authors don’t comment this? Please add a comment.
L.171: I suggest to use “significant or not significant differences” instead of “interactions”.
L.172: “The total FTMR ranged from…” specify what parameter the authors refer to.
It’ my opinion that the authors improperly use verbs as “interact”, “act”, “change” to describe the effects of the variables on the measured parameters. I suggest a revision.
Tables:
As for the text, I have the impression that also tables were not well prepared:
Titles are incomplete.
Footnotes refer to a coding of the treatments never used in the text (i.e.T1-T4).
Instead of “interaction” I suggest to write “Significance”, or “P”.
“Item” columns should align to the left, to make the text more clear.
In table 1, the chemical composition is expressed as % of DM, including the DM content, this is incorrect (I suggest to express the DM as g/kg, and the chemical composition of the DM as %DM). As it’s written, it seems that the pH is expressed as % of DM as well as the chemical composition of the feed. Cyanide probably should be substitute with HCN.
Author Response
Response: Thanks for the comments and now, we have improved our manuscript accordingly the comment made by the reviewers already. Please see more details in manuscript with track changes where we have revised or modified. Please see more response below.
Title: the title, although long, doesn’t include all the parameters analyzed; I assume that feed intake and digestibility are included in “feed utilization”, however is totally missing the analysis of rumen characteristics. I suggest to modify the title.
Response: We have modified to “Effect of sulfur levels in fermented total mixed ration containing fresh cassava root on feed utilization and rumen characteristics, microbial protein synthesis and blood metabolites in Thai native beef cattle”. Please see in manuscript.
Simple summary: There is space to improve it (max 200 words); I think is not very complete and clear, I suggest to improve it. As example, in this sentence “A high levels of HCN from fresh cassava root could be detoxified by a sulfur addition with an ensiling process to become nontoxic to cattle” the concept of detoxification is repeated twice, and it seems that the ensiling process is needed to achieve it (while, I don’t think it’s true).
Response: Thanks and we have modified as “Feeding of fresh cassava root to animals is restricted because it contains hydrocyanic acid at a high level, which is the origin for poisoning. A high levels of hydrocyanic acid from fresh cassava root could be detoxified by a sulfur addition to become nontoxic to cattle. The addition of 2% sulfur in fermented total mixed ration containing fresh cassava root and ensiling for 7 days could improve the dry matter digestibility, efficiency of microbial protein synthesis, total volatile fatty acid and propionic acid and blood thiocyanate concentrations.” Please see in manuscript.
Abstract: it is too long, the Instructions for the Authors clearly set a limit of 200 words, while the present abstract is of about 400. It needs to be reduced.
Response: We have modified to “The influence of sulfur included in fermented total mixed ration (FTMR) containing fresh cassava root on rumen characteristics, microbial protein synthesis and blood metabolites in cattle was evaluated. Four Thai native beef cattle were randomly assigned according to a 2x2 factorial in a 4 ×4 Latin square design, and dietary treatments were as follows: factor A included a level of sulfur at 1% and 2% in total mixed ration, and factor B featured ensiling times at 0 and 7 days. The digestibility of dry matter was increased when supplemented at 2% sulfur in the FTMR. Blood thiocyanate increased by 75.7% when supplemented at the 2% sulfur level with the ensiling time at 7 days compared to the 1% sulfur level and no ensiling (P<0.01). Bacterial populations were significantly different with an FTMR inclusion of sulfur at 2% and 7 days of ensiling. Furthermore, microbial crude protein and the efficiency of microbial protein synthesis were higher values with the FTMR containing the 2% sulfur level and being ensiled for 7 days (P<0.01). Thus, the high levels of hydrocyanic acid from fresh cassava root could be detoxified by a sulfur addition with an ensiling process to become nontoxic to cattle.” which consisted of 198 word. Please see in manuscript.
Introduction: generally, is complete and give an overall background on the topic. I have a specific remark on the hypothesis: from the introduction is clear that both the sulfur addition and the ensiling process reduce the HCN concentration in cassava root, so I would remove this from the hypothesis, and leave only the effect on feed utilization and rumen fermentation efficiency.
Response: Thanks and now we have modified as your comment as “It was hypothesized that sulfur addition and the ensiling method could improve feed utilization and rumen fermentation efficiency”. Please see in manuscript.
I also suggest to specify in the goal of the work that the breed chosen was a Thai native breed of cattle.
Response: Yes, we have specified as “Thus, goal of the work was to study the influence of sulfur doses included in FTMR containing fresh cassava root on nutrient intake, rumen characteristics, microbial protein synthesis and blood metabolites in Thai native breed of cattle.” Please see in manuscript.
Materials and methods:
Something very important, the Statistical analysis is totally missing, please provide information about this.
Response: We have added as “2.3 Calculations and statistical analysis
Data appropriate for the 2x2 Factorial in a 4× 4 Latin square design were analyzed sing GLM procedure (SAS, 1989). The model is following:
Yijk= μ+Mi+Ej+Ak+Pl+ εijk
where: Yijk, observation from cattlej, receiving diet i, in period k; μ, the overall mean, Mi, effect of the various doses sulfur (i= 1, 2), Ej, effect of the different ensiling time (j = 0,7), Ak, the effect of animal (k = 1, 2, 3, 4), Pl, the effect of period (l =1, 2, 3, 4), and εijk the residual effect. Present results as mean values with the standard error of the means. Differences between treatment means were tested by Duncan’s New Multiple Range Test, and differences amongst means were considered statistically significant at P<0.05were accepted as showing tendencies toward significance.” Please see in manuscript.
L.103: The description of the sampling protocol is very poor, please improve (e.g. were samples collected every day? Once a day? In the tables there is reference at 2 sampling times, but here nothing. Amount of sampling?)
Response: We have modified and indicated in section of “2.2. Sample collection and chemical analysis” as “…..Ten ml of blood were sampled from the animals’ jugular vein at 0 and 4 hours after feeding on the 21th day of each period to analyze BUN and blood thiocyanate, as described by Lambert et al. [18]. Approximately 45 ml of ruminal fluid samples were collected (also at 0 and 4 hours after feeding) on the 21th day of each period through a stomach tube connected to a vacuum pump….”Please see in manuscript.
L.85-86: the authors correctly state that the decision of the ensiling time was based on previous research, however they should do it also for the sulfur concentration, on what basis they choose 1 and 2%?
Response: Thanks, and we have provided already as “The optimum sulfur level and ensiling time were selected from our previous study [9] who indicated that it could improve kinetics of gas and nutrient digestibility while maintaining ruminal fermentation parameters and the rate of HCN disappearance.” Please see in manuscript.
L.89: the moisture content can’t be expressed as % of DM, considering that the DM doesn’t include the water.
Response: We have modified and remain only % as unit of moisture. Please see in manuscript.
L.90: what does it mean that FTMR was ensiled in plastic sealed containers to “ensure ingredient and chemical composition”? Please, make it more clear.
Response: We have modified as “…analyze ingredient and chemical composition”, please see in manuscript.
L.95: previously the animals were fed ad libitum, while in this phase at 90% of the previous feed intake; please, mention that the feed intake was previously measured.
Response: We have modified as “……. Each period ended with 7 days in which the animals were transferred to metabolism crates and fed the FTMR at 90% of the previous feed intake, during which all feces and urine were sampled. The feed intake was previously measured….” Please see in manuscript.
L.106: remove the organic matter from the chemical analysis; organic matter is obtained by calculation, not by chemical analysis.
Response: We have already removed from the Line. Please see in manuscript.
L.108: Please correct “Ata510”
Response: We have modified as “ At a 510”, please see in manuscript.
L.107,114: I don’t recommend to start a sentence with an abbreviation or symbol.
Response: We have modified, please see in manuscript.
L.120: specify what MCP is.
Response: We have modified, please see in manuscript.
L.134: Please, give more details on the protocols used, and some reference.
Response: We have modified, please see in manuscript.
Results and discussion:
without having the description of the statistical analysis is not very easy to understand and comment the results. In the tables the authors present the data at 0 and 4h after feeding, and the mean of these two sampling times, however in the materials and methods there is no description of this sampling protocol, and in the description of the results they don’t compare the two sampling times, but only the mean value. Please, justify this decision, and if they are not needed, I suggest to remove the values of the two sampling times from the tables.
Response: Thanks for suggestion, we have removed the date at 0 and 4 h from the tables and only show in mean values for blood metabolites and rumen characteristics. Thus, please see in tables.
Moreover, although for most of the parameters measured there is not effect of the interaction by the two variables (S*E), often the authors compare the values obtained with the 2% sulfur after 7d of ensiling with those obtained with the 1% sulfur at 0d of ensiling process. I believe this is incorrect, and can create confusion in understanding the results. I suggest to modify this approach.
Response: Thanks for suggestion. Now, we have revised according to your comments. In case there were no interaction between factor, it was separated described individual factor such pH value in subtopica of “Nutritional contents in the diets” and blood thiocynate in subtopic of “3.3. Ruminal fermentation, blood metabolites, microorganisms and purine derivatives”. However, some paprameter such as VFA which has interection between sulfue and ensiling time, then we were describes by the two variables (S*E) effect. Please see in manuscript.
L. 139-140: simplify this first sentence, a suggestion: “The chemical composition of the treatments is shown in Table 1”.
Response: Thanks for suggestion, we have modified and please see in manuscript.
I would separate the pH and the HCN from the chemical composition description.
Response: Thanks for suggestion, we have separated. In addition, we have recalculated percent reduction of HCN when 1 and 2% sulfur supplementation. Please see in manuscript.
L. 146: were lactic acid bacteria inoculated to the mixture before ensiling? Specify it in the materials and methods sections. The authors suggest that the lowest pH for the 2% sulfur treatment after 7 days of ensiling is due to the bacteria action during the ensiling process; this might be true, however why it’s not so for the 1% sulfur treatment?
Response: No, lactic acid bacteria did not inoculated to the mixture before ensiling and we have added into the section of material and methods, please see in manuscript. In addition, the lowest pH for the 2% sulfur treatment after 7 days of ensiling could be because increasing number of lactic acid bacteria during ensiling process of FTMR. The increasing of lactic acid bacteria population might be due to additional of high sulfur (2%), which sulfur is the one essential mineral for cell synthesis and maintenance of the cellular metabolism of microbes in the FTMR.
L.152-153: I don’t understand this sentence: “…, and it is attributed that this critical pH value varies with the DM content of the feed”; please clarify.
Response: We have modified the sentence and changed to “Even if the pH values of all FTMR with or without sulfur levels were not below 4.2, they were well preserved in anaerobic condition.” Please see in manuscript.
L.155-156: Also the other treatments reduced the HCN concentration more than 99% even without the ensiling process, why the authors don’t comment this? Please add a comment.
Response: Thanks for the comment on this point. The reduction of HCN concentration even without the ensiling process could be due to the preparation of TMR process by machine may generate high temperature, thereby resulting in lower HCN. We have given the comment in the text already.
L.171: I suggest to use “significant or not significant differences” instead of “interactions”.
Response: We have modified as “no interaction”, please see in manuscript.
L.172: “The total FTMR ranged from…” specify what parameter the authors refer to.
Response: We have modified as “The total FTMR intake were similar among feed groups and were ranged from 100.8 to 109.1 g/kg BW0.75.” Please see in manuscript.
It’ my opinion that the authors improperly use verbs as “interact”, “act”, “change” to describe the effects of the variables on the measured parameters. I suggest a revision.
Response: Thanks and we have changed throughout manuscript. Please see in the text.
Tables:
As for the text, I have the impression that also tables were not well prepared:
Response: We have modified the table according to the comment provide in the text section. Please see in manuscript.
Titles are incomplete.
Response: We have modified, please see in Tables.
Footnotes refer to a coding of the treatments never used in the text (i.e.T1-T4).
Response: We have removed from footnote, please see in Tables.
Instead of “interaction” I suggest to write “Significance”, or “P”.
Response: We have modified to “P value”. Please see in Tables.
“Item” columns should align to the left, to make the text more clear.
Response: Thanks for suggestion, however, this is the journal style which modified by Assistant Editor. Thus, we confirmed to use present form.

Reviewer 2 Report
I have some specific question below. Please answer.
L10; HCN --- What is this ?
L32; MCP --- What is this ?
L39; HCN --- What is this ?
L64-L72; supplementing fresh cassava root at 1.5% BW with a feed block containing 4% sulfur did not adversely affect
in vitro study 2% sulfur in TMR containing fresh cassava root fermented for 21 days could improve kinetics of gas and nutrient digestibility while maintaining ruminal fermentation parameters and the 54% rate of HCN disappearance
--- We have already known above things. Why did you conduct this study ? I think it is unnecessary. What is new ?
L93; proceeded for 4 21-day periods --- proceeded for 4 × 21-day periods
L108; Ata510 nm --- At a 510 nm
L114; N --- What is this ?
L118; W --- What is this ?
L121; What are small x ?
L125; animals’jugular --- animals’ jugular
L126; the 21st --- the 21th
BUN --- What is this ?
L132; VFA --- What is this ?
L133; HPLC --- What is this ?
L139; DM, OM, CP, NDF, ADF --- What are these ?
OM, ash --- Do you think it is necessary both ?
L145; with2% --- with 2%
L149; sufficient lactic acid and thus decreased pH when the ensiling time was extended 7 days --- Why did you use 7 days ? I feel it is short for ensilage. If you use more than 30 days for example, pH will decrease and be well fermented.
L149; extended7 days --- extended 7 days
L154; ensiling for7 days --- ensiling for 7 days
L201; NH3-N as an energy source --- I think NH3-N as material for cell synthesis and carbohydrate as an energy source. How do you think ?
L203; BUN was not altered --- How do you think about relationship with ruminal NH3-N ?
L223; inclusionof1% --- inclusion of 1%
L236; for7 days --- for 7 days
L252; from 105.8 to 113.8 mmol/l 64.6 to 66.6, 21.6 to 27.3, and 8.1 to 11.8 mol/100 mol --- Are these large differences ? Do these differences affect for whole body animal ?
L256; ensiling time improves the amount of available carbohydrates --- I think lactic acid bacteria use available carbohydrates during ensiling. How do you think ?
L258; shifting structural carbohydrates to starch --- I cannot understand. Please tell me.
L264; Supapong and Cherdthong [9] found that 2% sulfur levels of supplementation in TMR containing fresh cassava root increased propionic acid by 10.9% when compared to no sulfur being supplemented. --- I think it is enough this study. What is meaning of your study ? Anythig new ?
Table 1. DM or Dry matter ?
Organic matter, Ash --- Do you think it is necessary both ?
Fresh cassava root contain 53.1% NDF and 31.2% ADF. Is this OK ?
Footnote --- What are these ? Where are T1, T2, T3 and T4 ?
Table 2. Table 2. of FTMRon DM intake, --- Table 2. FTMR on DM intake,
DM or Dry matter ?
BW, ME, aNeutral detergent fiber --- What are these ?
Footnote --- Where are T1, T2, T3 and T4 ?
DM digestibility --- significant different
OM digestibility --- no different
What do you think this ? Did ash digestibility change ?
Table 3. of FTMR --- FTMR
You use "Rumen ecology" "Ruminal microbes, cell/ml". Where are blood parameters ?
Footnote --- Where are T1, T2, T3 and T4 ?
Table 4. of FTMR --- FTMR
Nitrogen or N ?
L247; (P<0.05).MCP:microbial crude protein. EMPS:efficiency --- (P<0.05). MCP: microbial crude protein. EMPS: efficiency
Author Response
Response: Thanks so much for your interested in our work and recommended for publication after major revision. Now, we have improved our manuscript accordingly the comment made by the reviewers already. Please see more details in manuscript.
L10; HCN --- What is this ?
Response: We have provided as “hydrocyanic acid”, please see in manuscript.
L32; MCP --- What is this ?
Response: We have provided as “microbial crude protein”, please see in manuscript.
L39; HCN --- What is this ?
Response: We have provided as “hydrocyanic acid”, please see in manuscript.
L64-L72; supplementing fresh cassava root at 1.5% BW with a feed block containing 4% sulfur did not adversely affect
in vitro study 2% sulfur in TMR containing fresh cassava root fermented for 21 days could improve kinetics of gas and nutrient digestibility while maintaining ruminal fermentation parameters and the 54% rate of HCN disappearance
--- We have already known above things. Why did you conduct this study ? I think it is unnecessary. What is new ?
Response: Right, previous reports on fresh cassava root have been elucidated, however, there are require new feed regimes and new novel present in current study.
“supplementing fresh cassava root at 1.5% BW with a feed block containing 4% sulfur did not adversely affect” ---this work very effective for strategic supplement feed block, however, there are some limitation such as feed block preparation (require machine), animal lick needed, and separate feed fresh cassava root. In case animal low intake of sulfur (low lick) from feed block, HCN may toxicity to animals. Thus, incorporate sulfur with fresh cassava root may support animal to detoxify HCN.
“in vitro study 2% sulfur in TMR containing fresh cassava root fermented for 21 days could improve kinetics of gas and nutrient digestibility while maintaining ruminal fermentation parameters and the 54% rate of HCN disappearance” ---This work was provided alternative feeding sulfur with fresh cassava root together without any effect from low sulfur intake and no need machine to make feed block. However, result from this study required to elucidate more in animal performance particularly beef cattle in order to provide information of HCN detoxify, feed utilization and rumen fermentation. Thus, the influence of sulfur doses included in FTMR containing fresh cassava root on nutrient intake, rumen characteristics, microbial protein synthesis and blood metabolites in Thai native breed of cattle is needed.
L93; proceeded for 4 21-day periods --- proceeded for 4 × 21-day periods
Response: We have modified as comment, please see in manuscript.
L108; Ata510 nm --- At a 510 nm
Response: We have modified as comment, please see in manuscript.
L114; N --- What is this ?
Response: It was nitrogen and we have modified, please see in manuscript.
L118; W --- What is this ?
Response: It was weigh of animal and we have modified, please see in manuscript.
L121; What are small x ?
Response: x mean multiplying of equation.
L125; animals’jugular --- animals’ jugular
Response: We have modified, please see in manuscript.
L126; the 21st --- the 21th
Response: We have modified, please see in manuscript.
BUN --- What is this ?
Response: It was blood urea nitrogen. We have modified, please see in manuscript.
L132; VFA --- What is this ?
Response: It was volatile fatty acid. We have modified, please see in manuscript.
L133; HPLC --- What is this ?
Response: It was “high-performance liquid chromatography”. We have modified, please see in manuscript.
L139; DM, OM, CP, NDF, ADF --- What are these ?
Response: We have explained in manuscript already, please see in Text.
OM, ash --- Do you think it is necessary both ?
Response: We have removed OM already, please see in manuscript.
L145; with2% --- with 2%
Response: We have modified, please see in manuscript.
L149; sufficient lactic acid and thus decreased pH when the ensiling time was extended 7 days --- Why did you use 7 days ? I feel it is short for ensilage. If you use more than 30 days for example, pH will decrease and be well fermented.
Response: Thanks for suggestion, in experiment use ensiling 7 days because treatment has supplement sulfur combine with high fresh cassava root in TMR. Sulfur an essential element to microorganisms synthetic and fresh cassava root as a non structural carbohydrate is beneficial in bacteria growth and could speed-up fermentation process of FTMR to rapidly decreased pH. In addition, ration of 40% roughage with high concentrate at 60% could be more supporting fermentation process, thereby resulting reduced pH. Moreover, based on our in vitro study confirmed that ensiling for 7 days could deceased pH as well.
L149; extended7 days --- extended 7 days
Response: We have modified, please see in manuscript.
L154; ensiling for7 days --- ensiling for 7 days
Response: We have modified, please see in manuscript.
L201; NH3-N as an energy source --- I think NH3-N as material for cell synthesis and carbohydrate as an energy source. How do you think ?
Response: Yes, we agreed and we have modified as your comment as “This could be due to ruminal microorganisms’ using NH3-N as material for cell synthesis and carbohydrate as an energy source .” Please see in manuscript.
L203; BUN was not altered --- How do you think about relationship with ruminal NH3-N ?
Response: Increasing of ruminal NH3-N but BUN was not altered have been observed from this study. This could be because ruminal NH3-N is high beneficial supply for microbial protein synthesis with carbohydrate from fresh cassava root, thereby resulting not altered BUN absorb to blood. This result was related to nitrogen utilization and microbial protein synthesis data. Thus, we have already added in section of “3.3. Ruminal fermentation, blood metabolites, microorganisms and purine derivatives” Please see in manuscript.
L223; inclusionof1% --- inclusion of 1%
Response: We have modified, please see in manuscript.
L236; for7 days --- for 7 days
Response: We have modified, please see in manuscript.
L252; from 105.8 to 113.8 mmol/l 64.6 to 66.6, 21.6 to 27.3, and 8.1 to 11.8 mol/100 mol --- Are these large differences ? Do these differences affect for whole body animal ?
Response: Yes, it should be affect for animal. Based on our study there were significantly highest in total VFA and propionate when animal fed FTMR (2% sulfur with 7 days ensiling). High total VFA and propionate could be provided high substrate synthesis energy and fat for animal and may affect for whole body animal. However, these findings should be further investigated regarding milk production to elucidate the effect of FTMR on animal production.
L256; ensiling time improves the amount of available carbohydrates --- I think lactic acid bacteria use available carbohydrates during ensiling. How do you think ?
Response: Yes, we agreed. However, due to high available carbohydrate source from fresh cassava root, it may remained available carbohydrate escape from lactic acid bacteria utilization and could be supply for VFA synthesis in the rumen.
L258; shifting structural carbohydrates to starch --- I cannot understand. Please tell me.
Response: We have modified as “In addition, a long ensiling time could provide a more carbohydrates and resulting in a high total VFA and propionic acid concentration in the rumen” Please see in manuscript.
L264; Supapong and Cherdthong [9] found that 2% sulfur levels of supplementation in TMR containing fresh cassava root increased propionic acid by 10.9% when compared to no sulfur being supplemented. --- I think it is enough this study. What is meaning of your study ? Anythig new ?
Response: Yes, we agreed. However, result from this study required to elucidate more in animal performance particularly beef cattle in order to provide information of HCN detoxify, feed utilization and rumen fermentation. Thus, the influence of sulfur doses included in FTMR containing fresh cassava root on nutrient intake, rumen characteristics, microbial protein synthesis and blood metabolites in Thai native breed of cattle is needed.
Table 1. DM or Dry matter ?
Response: Yes, it was “Dry matter”. We have modified, please see in manuscript.
Organic matter, Ash --- Do you think it is necessary both ?
Response: Yes, it should be expressed both values due to related with the calculation of digestibility and ME. In addition, there are many research papers in International Journal report both parameters.
Fresh cassava root contain 53.1% NDF and 31.2% ADF. Is this OK ?
Response: Yes, it is corrected.
Footnote --- What are these ? Where are T1, T2, T3 and T4 ?
Response: We have modified, please see in manuscript.
Table 2. Table 2. of FTMRon DM intake, --- Table 2. FTMR on DM intake,
Response: We have modified, please see in manuscript.
DM or Dry matter ?
Response: Yes. We have modified, please see in manuscript.
BW, ME, aNeutral detergent fiber --- What are these ?
Response: We have modified, please see in manuscript.
Footnote --- Where are T1, T2, T3 and T4 ?
Response: We have removed, please see in manuscript.
DM digestibility --- significant different
Response: Yes, we have check. It was different among treatments. It was small of SEM at 0.09.
OM digestibility --- no different
Response: Yes, we have check. It was no different among treatments.
What do you think this ? Did ash digestibility change ?
Response: No, it was no changed. Ash digestibility were 0.25-0.26% and differ only 0.01% DM.
Table 3. of FTMR --- FTMR
Response: We have modified, please see in manuscript.
You use "Rumen ecology" "Ruminal microbes, cell/ml". Where are blood parameters ?
Response: We have added “blood parameters” in table 3. Please see in manuscript.
Footnote --- Where are T1, T2, T3 and T4 ?
Response: We have modified, please see in manuscript.
Table 4. of FTMR --- FTMR
Response: We have modified, please see in manuscript.
Nitrogen or N ?
Response: We have modified, please see in Table 4.
L247; (P<0.05). MCP:microbial crude protein. EMPS:efficiency --- (P<0.05). MCP: microbial crude protein. EMPS: efficiency
Response: We have modified, please see in manuscript.
Thanks!

Round 2
Reviewer 1 Report
The comments and suggestions of the previous review have been accepted, and now the manuscript looks better, more clear, and more fluent to the reader.
Anyway, although I understad that the authors are not English native, I suggest a language review of the manuscript.
Few additional comments:
- table 4: footnotes incomplete, please check (the other tables are fine, maybe it's a copying/paste mistake).
- conclusion: there is a 4 lines sentence, is too long, I suggest to split it into 2 different sentences.
Author Response
Response to Reviewer: 1
The comments and suggestions of the previous review have been accepted, and now the manuscript looks better, more clear, and more fluent to the reader.
Response: Thank you very much and we have tried our best in order to make the reader clear.
Anyway, although I understad that the authors are not English native, I suggest a language review of the manuscript.
Response: Yes, we not English native speaker. Thus, before we submit paper we have send our manuscript to the company located in California United States, Order number 721850. Please see our receipt below.
Few additional comments:
- table 4: footnotes incomplete, please check (the other tables are fine, maybe it's a copying/paste mistake).
Response: We have changed to “S: P-value level of sulfur in diet. E: P-value ensiling times in diet. SEM: standard error of mean. a,b,cMeans within rows with different letters differ (P<0.05). MCP: microbial crude protein. EMPS: efficiency of microbial protein synthesis. N: Nitrogen.” Please see in Table 4.
- conclusion: there is a 4 lines sentence, is too long, I suggest to split it into 2 different sentences
Response: We have modified to “The addition of 2% sulfur in FTMR containing fresh cassava root and ensiling for 7 days could improve the dry matter digestibility, efficiency of microbial protein synthesis, total VFA and propionic acid and blood thiocyanate concentrations. In addition, the high levels of HCN from fresh cassava root could be detoxified by a sulfur addition with an ensiling process to become nontoxic to cattle. However, these findings should be further investigated regarding milk production to elucidate the effect of FTMR on animal milk production.” Please see in conclusion section.

Reviewer 2 Report
Table 2
I mean "DM digestibility was significant different, OM digestibility was no different, what do you think this ? Did ash digestibility change ?" I cannot understand relationship among DM, OM, and ash digestibility. Can you understand my question ?
Author Response
Response to Reviewer: 2
Table 2
I mean "DM digestibility was significant different, OM digestibility was no different, what do you think this ? Did ash digestibility change ?" I cannot understand relationship among DM, OM, and ash digestibility. Can you understand my question ?
Response: Thanks so much, we got your massage. Digestibility of DM was differed but not found in OM digestibility. This could be agree to your comment that ash digestibility may changed, however, present study did not determine digestibility of ash, thereby, it could not confirm your comments. In our opinion, increasing in ash digestibility could be occur when feed was fermented and lignin content can decrease. Thus, 7 days of ensiling and 2% sulfur might increase ahs digestibility.
Thank you.
